# The connection to the public's preferred sports analysis and physical education curriculum

Yong-Wook Kim[1], Jinyoung Han[2]*, Kyungtae Jang[3], Minsam Ko[1]*, Jaewoo Park[4], Seungyup Lim[5], Jin-Young Lee[4]

1 Department of Human-Computer Interaction, Hanyang University, Ansan, South Korea, 2 Department of Applied Artificial Intelligence, Sungkyunkwan University, Seoul, South Korea, 3 Department of Culture and Tourism, Komazawa Women's University, Tokyo, Japan, 4 Division of Sport Science, Hanyang University, Ansan, South Korea, 5 Division of Global Sport Studies, Korea University, Sejong, South Korea

* jinyounghan@skku.edu (JH); minsam@hanyang.ac.kr (MK)

**Data Availability Statement:** All relevant data are available on Zenodo: https://doi.org/10.5281/zenodo.6001490.

**Funding:** Ministry of Education of the Republic of Korea and the National Research Foundation of

## Abstract

People have their favorite type of sport, but such preferences tend to be shared for nearly a lifetime. How this preference persists remains inconclusive; hence, this study attempts to determine why people have different viewpoints on sports. It is reasonable to infer that these differences arise from differences in culture, occupation, and race. Therefore, we collected the following data and conducted research in Korea, the United States, and Japan, countries with various differences. The types of sports that people play were collected through surveys and comparisons among sports networks. Namely, "Sport Classification," "The K-12 Physical Education System (textbooks)," "Survey (actual physical activity)," "Simple Notification Service (SNS) Activity" have been examined to deduce the reason why any particular sport is played. Firstly, Korea, the United States, and Japan conduct different physical education courses. Hence, the results affect people's preferences. Secondly, what people post on SNS and their actual physical activities are different. Thirdly, the degree of connection between sports-type varied as well. Lastly, sports that serve the purpose of being regarded as hubs among sports-type were common in Korea, the United States, and Japan.

## Introduction

Michael Jordan, the greatest player in NBA, announced his retirement from professional basketball in October 1993. The retirement of the player considered as the "Greatest of All Time" in NBA history was unexpected. The next year, however, the world could witness him involved in another sport. He signed a contract with Chicago White Sox and spent the season as a baseball player in Birmingham Barons, the double-A team in MiLB. It was an unimaginable event where an athlete played both professional basketball and baseball, two seemingly unrelated sports. Ultimately, Jordan finished the 1994 season with a 0.202 batting average, 3 home runs, 51 runs batted in, 0.289 On Base Percentage, and .556 On Base Percentage Plus Slugging Percentage in the 127 games with double-A [1].

Korea (NRF-2020S1A5B8104091; https://www.nrf.re.kr; provided to M.K.); Technology Advancement Research Program funded by Ministry of Land, Infrastructure and Transport of Korean government.(21CTAP- C152247-03; https://www.kaia.re.kr/; provided to M.K.) The funders had no role in study design, data collection and analysis, decision to publish, or preparation of the manuscript.

**Competing interests:** The authors have declared that no competing interests exist.

However, the event demonstrated that a player with excellent physical ability could gain a prime opportunity by transitioning to another sport. Unlike Jordan, there is an athlete who succeeded in sports career transition. He is SeungHoon Lee, who acquired a gold medal in speed skating at the 2018 Winter Olympics–Men's Mass Start. He also won a gold medal in the 2008 World Short Track Speed Skating Championships– 3000m [2]. Although speed skating and short track share a common feature in that the players skate on ice, they are different sports in terms of their dissimilar stadium sizes and skating postures. In comparison with track-and-field, it can be similar to the situation where a sprinter challenges a marathon runner. Nonetheless, it is amazing to witness the world's best performances equally in such distinctly dissimilar sports.

Some cases of famous sports stars are as follows. Watching Rafael Nadal, the tennis player, and Cristiano Ronaldo, the football player, playing tennis with football, it was evident in the performance that Rafael Nadal has a remarkable ability in football. It is said that he was a football player in his childhood and a fan of Real Madrid, the professional football team in La Liga [3,4]. Moreover, Tiger Woods, the golf player, is known as the best friend of Serena Williams, the tennis player. For that reason, it is not awkward to witness him on the spectator's seat of a tennis court at the US Open, and his tennis ability is believed to be above amateur's level [5]. It is widely known that many players enjoy not only their main sports but also other types of sports. However, it is often mistaken to think that players are not able to learn the sports apart from their main ones and even afford to enjoy them. The mistake is often derived from a thought that if a player wishes to be the greatest in the world, it is necessary to conduct the training related only to his or her main sport. The fact is that the world's best players are enjoying multiple sports as hobbies. In other words, it can be understood that athletes enjoy multiple sports rather than only one sport.

If so, which patterns govern how people enjoy sports? Which sports do people, who normally enjoy baseball, play additionally? Do people enjoy golf like tennis? If people enjoying golf choose to play tennis, what connection leads people to make such a choice? Does the choice result from the experience of the K-12 Physical Education System? To find answers to these questions, such notions should be evaluated based on people's experiences. Sport types are important because they affect lifelong exercise activities in the school curriculum where exercise is first learned. Therefore, we analyzed the K-12 Physical Education System to see how people's learning experiences affect their lifelong exercise activities.

## Literature review

Spectator sports, which are popular among enthusiasts of professional sports and participation sports, differ in terms of leisure and life satisfaction [6–9]. Concerning spectator sports, football is popular in the USA, Korea, and Japan, countries known to prefer baseball [10]. Moreover, the result of the survey on people's physical activities in Korea, Japan, and the USA is illustrated in Table 1, representing the varying preferences of sports in each nation. Upon close examination, it is revealed that people in Korea like walking (40.8%), mountain climbing (23.2%), and fitness (11.9%). In contrast, data from Japan denotes a preference towards walking (57.0%), fitness (12.9%), and gymnastics (12.4%). Further, fitness sports (66.0%), outdoor sports (59.2%), and individual sports (45.3%) were ranked in ascending order in the USA (Table 1). The difference in the preferences of sports across countries is caused by various factors, such as culture and geography [10]. One of the main factors accounting for the differences can be the K-12 Physical Education System. This correlation is assumed because the K-12 Physical Education Systems in Korea, the USA, and Japan are different as adults tend to play accustomed sports through the aforementioned system [11].

**Table 1. Popularity of community sports in Korea, Japan, and the USA.**

| Sports | Korea [12] | Japan [13] | Sports | USA [14] |
|---|---|---|---|---|
| Aerobics | 2.7% | 0.1% | fitness sports | 66.0% |
| Badminton | 7.8% | 3.1% | | |
| Baseball | 2.3% | 5.9% | | |
| Basketball | 5.3% | 1.7% | individual sports | 45.3% |
| Billiards | 8.3% | 0.1% | | |
| Bowling | 7.1% | 4.7% | | |
| Cycling | 8.9% | 10.9% | outdoor sports | 59.2% |
| Dancing | 1.4% | 2.5% | | |
| Fencing | 0.7% | 0.1% | | |
| Fishing | 4.1% | 4.5% | racquet sports | 13.0% |
| Fitness | 11.9% | 12.9% | | |
| Football | 8.9% | 4.05% | | |
| Golf | 4.0% | 11.0% | team sports | 22.6% |
| Gymnastics | 9.4% | 12.4% | | |
| Martial arts | 1.7% | 0.1% | | |
| Mountain climbing | 23.2% | 3.9% | water sports | 13.7% |
| Ping pong | 3.5% | 3.2% | | |
| Rope skipping | 7.7% | 2.2% | | |
| Swimming | 8.3% | 5.2% | | |
| Tennis | 1.4% | 3.8% | | |
| Track and field | 3.3% | 12.2% | | |
| Volleyball | 1.5% | 1.9% | winter sports | 7.1% |
| Walking | 40.8% | 57.0% | | |
| Yoga | 6.3% | 6.3% | | |

Ministry of Culture, Sports and Tourism-Korea. 2018 [12]; Physical Activity Council, 2019 [14]; Japan Sports Agency, 2019 [13].

There are two main ways to select sports that are covered in the school curriculum. First, there is a standard for selecting basic sports for the physical development of students. Students have different athletic abilities depending on their physical age. Basic exercises that meet each age standard should be performed. These exercises help students develop their bodies. Second, there is a standard for selecting various sports to attract students' interest. Basic events such as running and walking are difficult to attract students' interest. It is difficult to create a sense of cooperation that is the basis of social life. Therefore, to integrate cooperation into physical education, various sports, such as group sports, should be utilized to attract students' interest. The physical education curriculum should consist of 'Basic exercises' and 'Fun sports' in harmony. These two conditions are contrary to each other. How the two are harmoniously distributed leads to either the success the or failure of physical education.

The various sports constituting the K-12 Physical Education System are integral because they influence individuals' lifelong sports activities. The emphasis on the physical education system in each country is stated below. The K-12 Physical Education System in Korea is based on body movement, which consists of health management ability, physical training ability, competition performance ability, and body expression ability. Through this process, participants are expected to learn various sports [15]. Japan's K-12 Physical Education System aims to improve health and physical strength by understanding basic movements of exercise and solving fundamental tasks [16–18]. The K-12 Physical Education System in the USA aims to improve various skills, knowledge, social behavior, and recognition of the value of actual

physical activity for health, enjoyment, challenge, self-expression, and/or social interaction [19].

As mentioned above, studies on the influence of the K-12 Physical Education System on lifelong sports activities have been reported in various ways [20,21]. A study demonstrating that differences in culture, geography, and economy has impact on sports activities was also reported [11,22]. Various studies have been conducted on the recognition of products and consumers in the marketing research field of public goods. In determining a location for a store and displaying products, the relationship between selling products and customers' preferences has been investigated in locations such as department stores [23,24].

Thus, to explain the sports network, which is the objective of this study, connections between sports should be analyzed. Studies on the analysis of correlations among sports are rare except in research that classifies and categorizes the characteristics of sports and then determines their positions [25,26]. For example, a variety of factors can be correlated, such as popularity, stadium, strategies, and players' movement skills. For an accurate analysis of correlations among sports, it is necessary to draw various sports networks, decipher their meanings, and explain them with great insight. It is also imperative to adapt to a means of network analysis that thoroughly examines the K-12 Physical Education System as well as the actual physical activity and Simple Notification Service (SNS) activities required to conduct such a study.

Generally, social network analysis can be explained by identifying the mutual connections between the subjects in the study group [27]. The connection mentioned here can be classified largely into three categories–"human vs. human," "object vs. object," and "event vs. event" [28]. The way to intuitively identify these connections is to visualize and observe the patterns of the connections. The visualized outcomes are composed of nodes and edges. A node refers to the subjects of these connections, such as humans, events, and objects. Edge happens to be a connection of the Node, which can have either directivity or only a simple connection without directivity according to the properties of the connections. We can intuitively perceive whether our relationship with others is good, bad, hostile, or intimate. As the perception is enabled by our experience, it has been regarded as unquantifiable. Nevertheless, if it is connected with the directivity-given edge, the connected networks can visualize mutual relations such as those among groups, organizations, and events. Therefore, the characteristics of the networks can be established. Besides, the deduction of properties and meanings of the networks can also be enabled by identifying the measured outcome in quantified figures [29,30].

## Materials and methods

This study method includes a survey of Koreans, Japanese, and Americans. The contents of the survey include the subject's physical activity and preference for sports events. The number of survey subjects was set at 350 each according to the survey method of previous study [31]. In addition, detailed explanations of the survey contents were notified to the survey subjects and consent was obtained for the survey. This study was conducted with the approval of IRB.

This study aims to analyze the correlations of sport types. The sports types that people play have been collected by surveys and the comparisons between sports networks–"Sport Classification," "The K-12 Physical Education System," "Survey (actual physical activity)," and "SNS Activity" to deduce the reason why they play the sport. Based on the connections between sports types constructed in each field, the associative sports networks were drawn. This study also verifies the characteristics of sports networks by investigating their similarities and examining whether they vary by country (i.e., Korea, the USA, and Japan). Notably, Korea and Japan were selected due to their close regional and cultural similarities, and the United States was selected due to its great influence on the political culture of Korea and Japan after World War II.

To achieve the goal of this study, data were collected through the process described below. A total of 24 sports types were selected as subjects for the analysis before data collection: aerobics, badminton, baseball, basketball, billiards, bowling, cycling, dancing, fencing, fishing, fitness, football, golf, gymnastics, martial arts, mountain climbing, ping pong, skipping rope, swimming, tennis, track and field, volleyball, walking, and yoga. The selection of the sports types was based on "A Survey on the Participation in Sports Activities in Korea," which was conducted through an expert meeting (i.e., five scholars from the discipline of sport studies).

First, the "Sport Classification" was carried out through expert surveys. The experts who participated in the survey were a group of six people with Ph.D degrees in sports. The expert group held three meetings over the course of two months to select survey items. The 41 categories (Table 2) classified by sports characteristics were selected during the experts' meeting, and the data were collected after receiving the response from the experts to identify the connections among sports types. The collected data was arranged multidimensionally according to 41 characteristics distributed among 276 pairs of Sport types, and the distance between the types

**Table 2. Sports characteristics.**

| Sports Characteristics |
|---|
| 1. Is the main exercise place indoors? |
| 2. Do you use limited venues (fields)? |
| 3. Are the sports venues being moved? |
| 4. Do we follow a set course when moving? |
| 5. Do you usually use your hands? |
| 6. Do you usually use your feet? |
| 7. Do you use your whole body? |
| 8. Do you use running motions? |
| 9. Do you use walking motions? |
| 10. Do you use rolling motions? |
| 11. Do you use body rotation? |
| 12. Do you use the arm rotation movement? |
| 13. Do you use the throwing motion? |
| 14. Do you use the hanging motion? |
| 15. Do you use jump actions? |
| 16. Do you use dribble movements? |
| 17. Do you use head movements? |
| 18. Do you use the ball rolling motion? |
| 19. Do you use the kick motion to kick the ball? |
| 20. Do you use your hand to receive the ball? |
| 21. Do you use hand-punching movements? |
| 22. Do you use equipment to hit the ball? |
| 23. Do you use the wielding motion of the equipment in your hand? |
| 24. Does it involve frequent physical contact with the other party? |
| 25. Do you use actions that pose threats to the other party? |
| 26. Is it an exercise that you usually do alone? |
| 27. Is it mainly a team sport? |
| 28. Is it an exercise that has an opponent? |
| 29. Is it an exercise that usually records scores? |
| 30. Is it primarily a time-measuring exercise? |
| 31. Is it an exercise that uses equipment or instruments a lot? |
| 32. Is it an exercise that usually uses a ball? |
| 33. Is it an exercise that requires wearing protective gear to protect the body? |
| 34. Is it an exercise that requires sports equipment? |
| 35. Is it an exercise that uses goalposts? |
| 36. Is it an exercise that uses a net? |
| 37. Is it an exercise that mainly uses hand-held equipment? |
| 38. Is it an aerobic exercise? |
| 39. Is it a muscle workout? |
| 40. Do you need music when you exercise? |
| 41. Is equipment essential to partake in the exercise? |

**Table 3. Textbooks.**

| Country | K-12 Physical Education Systems | | | |
|---------|---------|------|------------|-------|
| | Textbooks | Page | Paragraphs | Words |
| Korea | 2015 Physical Education Course [15] | 124 | 934 | 10,303 |
| USA | National Standards & Grade-Level Outcomes for K–12 Physical Education [19] | 913 | 2,134 | 26,637 |
| Japan | Elementary School Physical Education Course [16] Middle School Physical Education Course [17] High School Courses Physical Education Course [18] | 136 | 4,674 | 50,174 |

Ministry of Education-Korea, 2015 [15]; Couturier L, Stevie C, Shirley HH, 2014 [19]; Ministry of Education, Culture, Sports, Science and Technology-Japan, 2017a [16]; Ministry of Education, Culture, Sports, Science and Technology-Japan, 2017b [17]; Ministry of Education, Culture, Sports, Science and Technology-Japan, 2017c [18].

was measured to identify the connections between the 24 types. Responses were provided on a 5-point Likert scale. In addition, the distance between the types was measured using Jaccard's coefficient. Cronbach's alpha value was checked to verify the reliability of all 41 questions, and values between 0.822 and 0.929 were found for all questions.

For the analysis of the K-12 Physical Education System, the documents of the curricula in Korea, the USA, and Japan were collected as exhibited in Table 3. The documents of the curriculum in Korea include 10,303 words and 124 pages, and the ones in the USA comprise 50,174 words and 136 pages. Japan's curriculum includes 26,637 words and 913 pages. The paragraphs in each document were extracted to identify the connections between sports types that appeared in each curriculum. The numbers of paragraphs are classified as 934 paragraphs in Korea, 4,674 paragraphs in the USA, and 2,134 paragraphs in Japan. The sports types are connected through their classification into pairs of sport types that appear in the same paragraph.

Third, surveys (actual physical activity) were conducted in Korea, the USA, and Japan to identify the types of sports that people actually play. Third, surveys (actual physical activity) were conducted in Korea, the USA, and Japan to identify the types of sports that people actually play. A total of 1,662 people participated in the survey, with 547 people from the United States, 753 people from Korea, and 362 people from Japan participating. The surveys were conducted simultaneously in the three countries over the course of one month in May 2019. Convenience sampling was utilized as the survey method. Face-to-face surveys were adapted in Korea and Japan, while an online survey was used in the USA. The general features of the collected data are shown in Table 4. The connections between sports types were drawn by receiving responses to the survey questions inquiring about the sports played together. If two physical activities were responded to, they were connected once, and if three answers were retrieved, they were connected after being classified into each pair of sport types. The survey results verified that the number of people participating in more than two physical activities was 264 (60.55%) in Korea, 505 (78.54%) in the USA, and 172 (55.66%) in Japan. The relationship between occupation, age, income level, and sports events collected in the survey will be further analyzed in a follow-up study that has been developed.

Finally, the posting activities of the SNS were used for data analysis, wherein the collected posts related to physical activities were uploaded on Instagram. The collection was facilitated by web crawling using keywords related to 24 sports types on Instagram. Web crawling is a method of automatically collecting information posted online by creating an Internet bot program. Through this process, sports types were collected as data from posts uploaded by SNS users. The collected data are presented in Table 5. The connections between the sports types in the SNS activities were enabled by searching for posts written by the same SNS user. If two

**Table 4. Sample characteristic (actual physical activity).**

| Sample Characteristic | | Korea | USA | Japan |
|---|---|---|---|---|
| Gender | Male | 248 | 307 | 178 |
| | Female | 299 | 446 | 184 |
| Job Classification | Student | 263 | 28 | 152 |
| | Businessman | 129 | 37 | 102 |
| | Public servant | 2 | 63 | 1 |
| | Management | 10 | 103 | 11 |
| | Professional | 40 | 247 | 34 |
| | Self-employed | 32 | 117 | 27 |
| | Homemaker | 42 | 69 | 17 |
| | Others | 29 | 89 | 18 |
| Age Group | 21~30 years | 315 | 161 | 148 |
| | 31~40 years | 110 | 242 | 102 |
| | 41~50 years | 74 | 165 | 68 |
| | 51~60 years | 40 | 108 | 30 |
| | Over 60 years | 8 | 77 | 14 |
| Monthly Household Income | under 2,000 dollars | 176 | 164 | 124 |
| | 2,001~4,000 dollars | 163 | 283 | 102 |
| | 4,001~6,000 dollars | 92 | 174 | 48 |
| | 6,001 dollars or more | 116 | 132 | 88 |
| Exercise Frequency (per Week) | 1 time | 94 | 31 | 71 |
| | 2 times | 56 | 93 | 58 |
| | 3 times | 56 | 176 | 39 |
| | 4 or more times | 74 | 288 | 49 |
| | I don't exercise. | 111 | 110 | 53 |
| | I exercise irregularly. | 156 | 55 | 92 |
| Exercise Frequency (per session) | Less than 30 minutes | 120 | 152 | 77 |
| | 30~60 minutes | 146 | 370 | 88 |
| | 60~120 minutes | 65 | 94 | 51 |
| | Over than 120 minutes | 24 | 8 | 24 |
| | I don't exercise. | 111 | 110 | 53 |
| | Different from time to time | 81 | 19 | 69 |
| The types of Physical Activity You Do | I don't exercise. | 111 | 110 | 53 |
| | 1 type | 172 | 138 | 137 |
| | 2 types | 110 | 169 | 88 |
| | 3 types | 94 | 150 | 61 |
| | 4 or more types | 60 | 186 | 23 |
| Total | | 547 | 753 | 362 |

physical activities were responded to, they were connected once, and if more than three activities were undertaken, they were connected after being classified into each pair of sports types.

The collected data displayed above draws sports networks to identify the relations between sports. We introduce the notion of the "Sports Network" as an undirected graph G = (N, E), where a node indicates a sport and an edge indicates the relation between two sports. To see the similarity between "Sports Networks," both "Edge Overlaps" and "Node Overlaps" were measured. "Edge Overlaps" was measured using the method of "Jaccard's coefficient" and the method of "Pearson's correlation coefficient" was used to measure "Node Overlaps." While examining "Edge Overlaps" and "Node Overlaps" during the network analysis, the similarity

**Table 5. SNS (Instagram).**

| Sports | Korean | English | Japanese |
|---|---|---|---|
| Aerobics | 70,510 | 99,067 | 10,249 |
| Badminton | 120,352 | 120,477 | 78,324 |
| Baseball | 92,352 | 82,601 | 71,188 |
| Basketball | 102,555 | 80,570 | 85,220 |
| Billiards | 96,122 | 93,982 | 75,018 |
| Bowling | 99,954 | 75,142 | 58,415 |
| Cycling | 130,627 | 89,122 | 71,998 |
| Dancing | 40,892 | 78,057 | 102,359 |
| Fencing | 191,595 | 74,955 | 130,521 |
| Fishing | 117,441 | 68,557 | 74,811 |
| Fitness | 146,926 | 84,322 | 112,573 |
| Football | 82,275 | 89,616 | 79,598 |
| Golf | 140,838 | 79,074 | 84,726 |
| Gymnastics | 167,595 | 76,228 | 104,068 |
| Martial arts | 101,592 | 77,024 | 161,169 |
| Mountain climbing | 110,028 | 92,609 | 84,705 |
| Ping pong | 97,942 | 85,236 | 79,078 |
| Rope skipping | 121,717 | 92,353 | 15,997 |
| Swimming | 115,560 | 67,610 | 91,642 |
| Tennis | 118,433 | 79,541 | 84,745 |
| Track and field | 111,066 | 171,814 | 97,354 |
| Volleyball | 98,997 | 91,415 | 82,336 |
| Walking | 112,044 | 67,415 | 65,907 |
| Yoga | 156,496 | 105,706 | 144,332 |
| Total | 2,743,909 | 2,122,493 | 2,046,326 |
| | 6,912,728 | | |

between the two networks can be identified in figures. The differences recognized intuitively in the network graphs can be seen quantitatively through "Edge Overlaps" and "Node Overlaps" [32–34]. This study uses "R version 3.6.0," a statistical program, to analyze the data and "Gephi 0.9.2" to visualize the graphs.

## Results and discussion

### Textbook-oriented differences according to the countries

The comparative analysis that draws sports networks of "The K-12 Physical Education System," "Survey (actual physical activity)," and "SNS Activity" was carried out to see the differences in Korea, the USA, and Japan. The sports networks of the K-12 Physical Education System in each country are shown in Fig 1. As depicted in the graph, it is noticeable that the countries have distinct networks. In Table 6 where the detailed comparisons between the countries are highlighted, the rate of the similarity of "the K-12 Physical Education System" in Korea and Japan is evaluated as low, with 'Node Overlaps' of 0.349***. Moreover, the USA and Japan have a low rate of similarity indicated by the 'Node Overlaps' of 0.134*. Lastly, Korea and the USA are verified to have no similarity. Contrariwise, the similarity between Korea and Japan was thought to be due to geographical and cultural similarities, as well as their curricula, which are dominantly run by the government. The results derived from the USA, on the other hand, are considered to be caused by the fact that only brief guidelines for the curriculum are

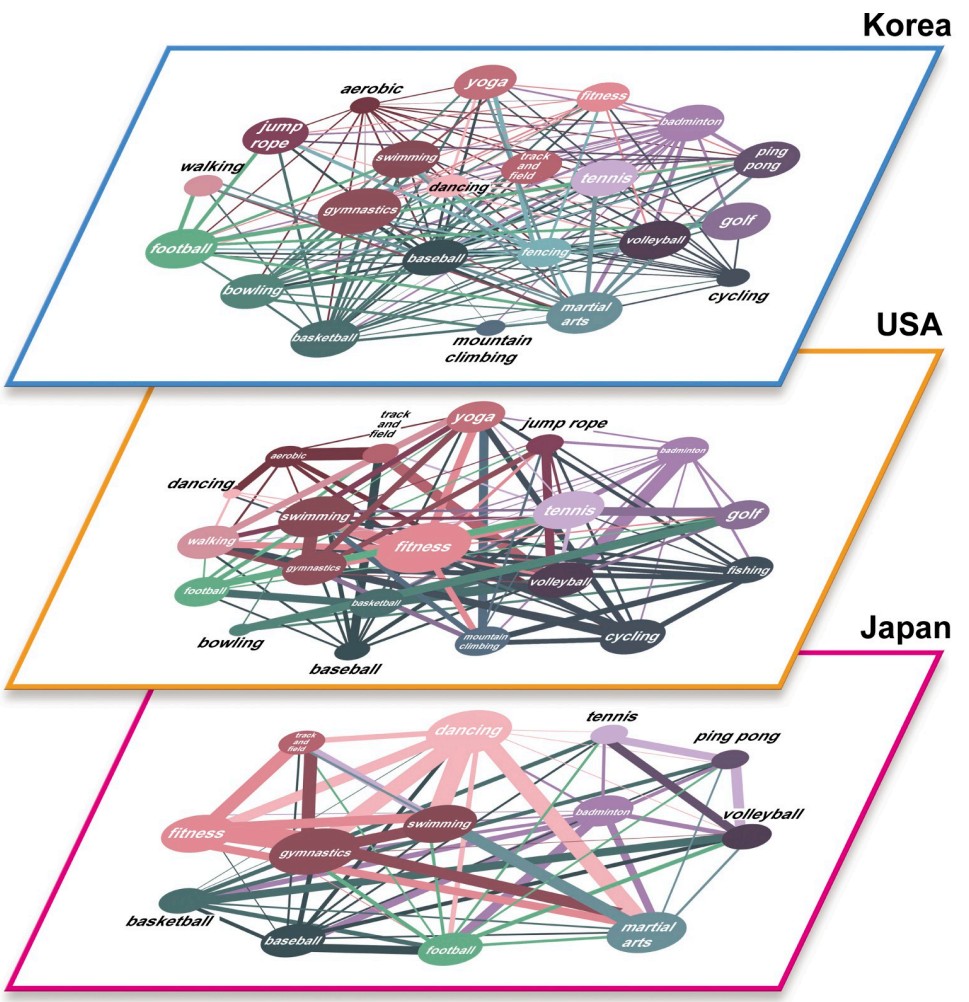

**Fig 1. Textbooks.**

suggested in there [35,36]. Consequently, the curricula in Korea and Japan have no impact on actual physical activities, while the USA does denote that capability to influence certain elements. This effect persists because of the differences in each country's curricula.

## Connections between textbooks and actual physical activity

The differences are evident when comparing the results of 'The K-12 Physical Education System" and "Survey (actual physical activity)" in each country. Specifically, for the USA, a low

**Table 6. Network overlaps (textbook vs. country-by-country).**

| X | Y | Edge Overlaps (Jaccard's coefficient) | Node Overlaps (Pearson's Correlation coefficient) |
|---|---|---|---|
| Korea textbook | USA textbook | 0.10840738 | 0.09728171 |
| Korea textbook | Japan textbook | 0.080288147 | 0.349266*** |
| USA textbook | Japan textbook | 0.034182492 | 0.1346034* |

* $p < .05$

** $p < .01$

*** $p < .001$.

**Table 7. Network overlaps.**

| X | Y | Edge Overlaps (Jaccard's coefficient) | Node Overlaps (Pearson's Correlation Coefficient) |
|---|---|---|---|
| Korea textbook | Korea survey | 0.119293298 | -0.009021564 |
| USA textbook | USA survey | 0.11139786 | 0.2983137*** |
| Japan textbook | Japan survey | 0.054412987 | 0.0001298196 |

* $p < .05$, ** $p < .01$

*** $p < .001$.

similarity is indicated, which is 0.298*** , as revealed in Table 7. Nonetheless, no similarity appears between Korea and Japan. Consequently, the sports networks of "The K-12 Physical Education System" and "Survey (actual physical activity)" are shown to be different in all three countries, Korea, the USA, and Japan. This dissimilarity means that the sports activities dealt with in "The K-12 Physical Education System" did not mimic actual sports activities.

Education Systems do not influence the sports activities of adults. Of course, this notion does not signify that there is no use in learning sports that are not enjoyed in everyday life. "The K-12 Physical Education System" consists of curricula regarding the effects of the educational process of students' physical and psychological developments [37–40]. Accordingly, there is an aspect of the system that intentionally organizes and teaches sports that are difficult to be enjoyed in actual life. It is, however, discovered that there is no relation between the K-12 Physical Education System and actual situations for the activities. This disassociation further suggests that the sports dealt in "The K-12 Physical Education System" were selected more carefully and on scientific grounds (Fig 2).

## SNS is different from the actual physical activity

The results of the surveys on actual physical activities and the search for activities-related posts on the Internet are highlighted in Fig 3 and Table 8. As shown above, the differences in Korea, the USA, and Japan are all recognized. Node overlaps of Korea and USA show low rates of similarity, presenting 0.219*** and 0.254*** , respectively. In the case of Japan, noticeable similarity cannot be determined with a value of 0.114. The gap between SNS posts and the results of the survey on actual physical activities is thought to be a result of the tendency of SNS activities to boast about one's activities to acquaintances or unspecified individuals [41]. Therefore, "walking" was not frequently posted in the SNS because it is not thought to have much ability to boast about. On the other hand, cycling and fitness appeared the most in the SNS activities. This occurrence is due to the characteristic of SNS, which can accumulate many reactions to the posts that are fancy and worthy of attention [42]. Hence, the difference between SNS and actual physical activity cannot easily be enjoyed in real life; however, posts that boast about privileges tend dominate.

## Sport classification is rarely correlated in the three fields

The correlation of Sport Classification is shown in Fig 4. The two sports with the highest similarity are tennis and badminton. While the ranking of the connection between the two is measured as 1 in the sports types, it is measured as low as 204 (Korea), 20 (USA), and 39 (Japan) in textbooks. Besides, it is also very low in the surveys, as in 123 (Korea), 141 (USA), and 139 (Japan). Lastly, the results of the SNS activities were 27 (Korea), 55 (USA), and 113 (Japan). Consequently, the correlations of Sport Classification are considered to have no similarities with all the studied fields, such as educational activities, actual physical activities, and SNS

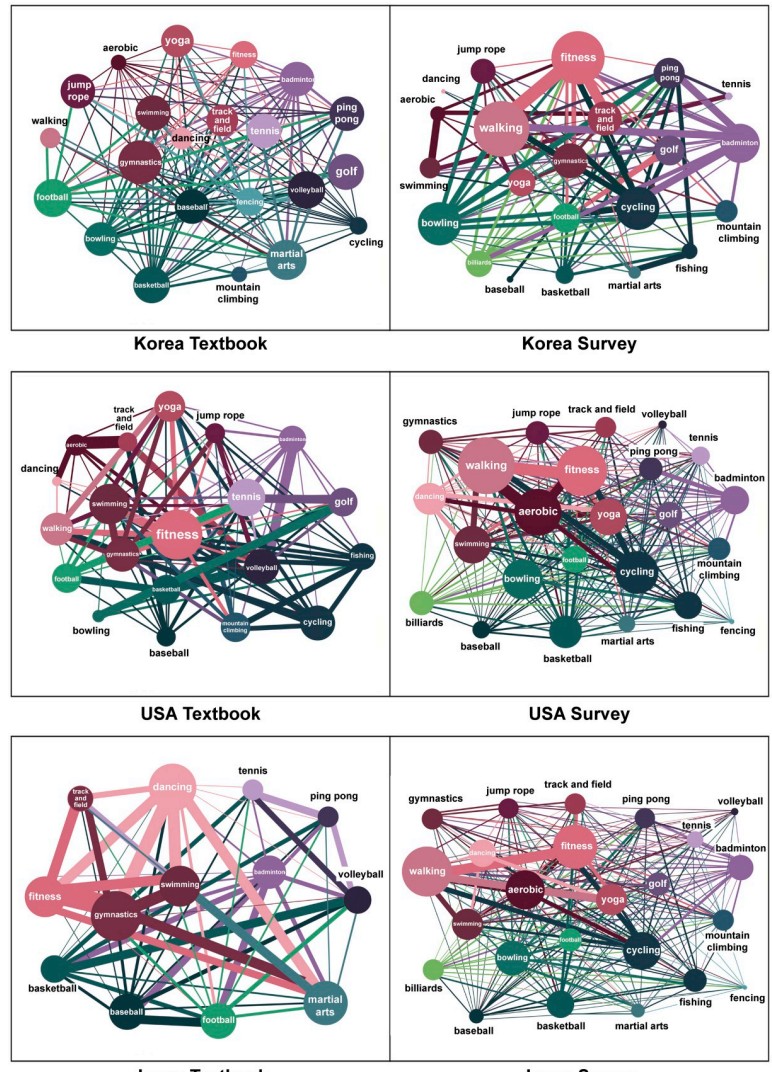

**Fig 2. Textbooks and survey (actual physical activity).**

activities (Table 9). The similarities between sports types classified by the expert group were confirmed by measuring various characteristics of the sports types. Thus, ping pong and tennis have a similar relationship, as do tennis and badminton, but the results from the three fields were different. Therefore, it may seem that the similarities between the sports types have no special meaning. However, as the connections quantify the similarities between sports types, it is meaningful to select sports with no similarities for a curriculum that encourages various physical activities.

## Top linked sports types

The sports that present the highest relations in each field–"The K-12 Physical Education System", "Survey (actual physical activity)," and "SNS activities" are seen as follows. First, the high relations in educational activities are as illustrated in Fig 5. Sports with high relations are shown equally in basic sports, such as walking, track and field, fitness, and gymnastics in Korea, the USA, and Japan. Looking into the distinct features of each country, it is noticeable

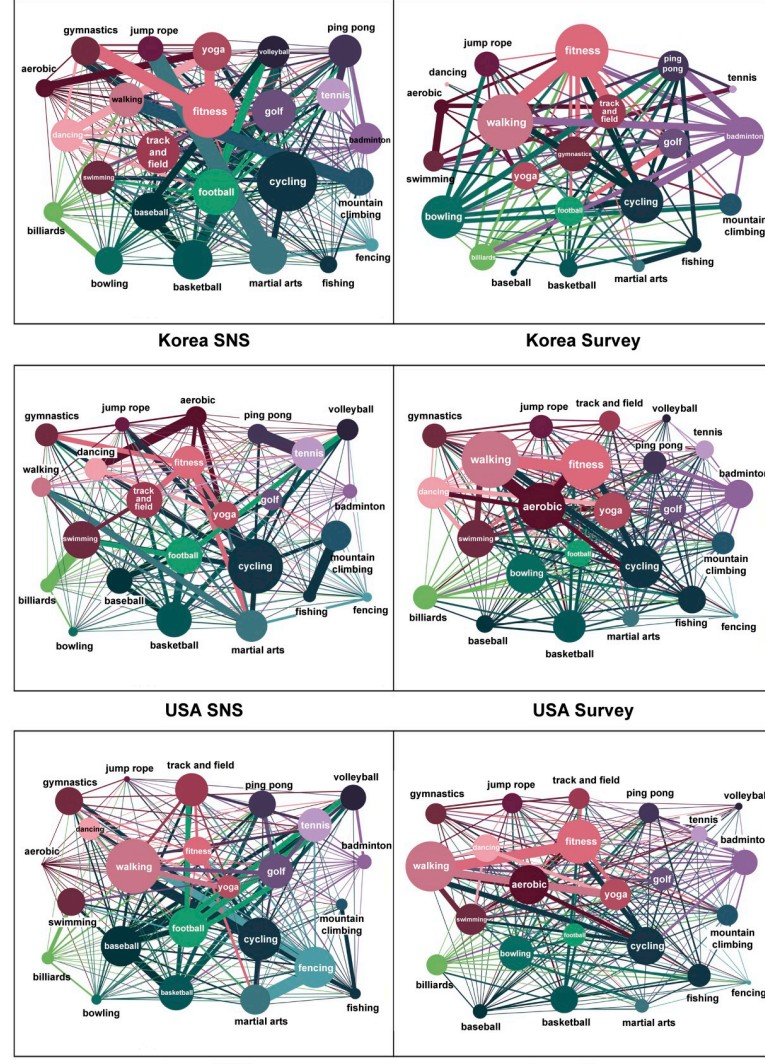

**Fig 3. SNS and survey (actual physical activity).**

that martial arts are practiced in both Korea and Japan, but not in the USA. According to a study analyzing the perception of martial arts, individuals in Korea and Japan are reported to engage in martial arts for their safety and physical and mental discipline. In particular, as the positive influences on adolescence have been recognized, many adolescents appear to be avid

**Table 8. Network overlaps (SNS vs. survey).**

| X | Y | Edge Overlaps (Jaccard's coefficient) | Node Overlaps (Pearson's Correlation coefficient) |
|---|---|---|---|
| Korean SNS | Korea survey | 0.138718944 | 0.2192403*** |
| English SNS | USA survey | 0.25946794 | 0.2542677*** |
| Japanese SNS | Japan survey | 0.246101785 | 0.1143671 |

* $p < .05$, ** $p < .01$

*** $p < .001$

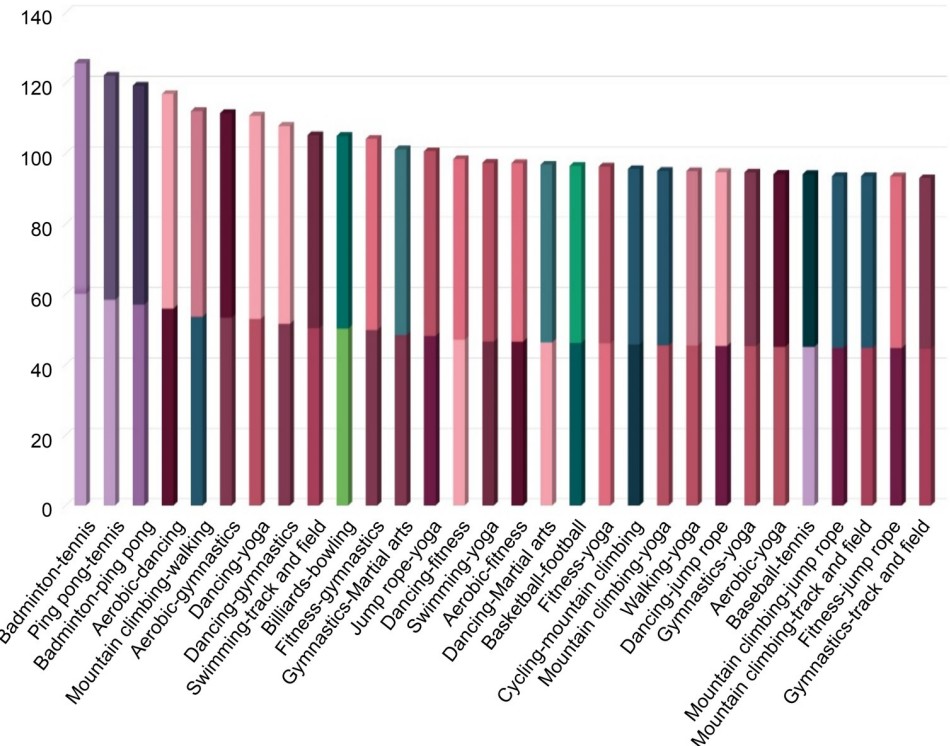

**Fig 4. Sport classification—Top 30.**

practitioners of martial arts [43]. Martial arts, however, have been popularized in the USA as being centered on people with special goals, such as attaining a certain degree of physical fitness [44]. Therefore, it seems that this knowledge is not applied to educational activities owing to the belief that the practice can transform adolescents' bodies into weapons and lead to serious violence.

Secondly, the relations indicated in actual physical activities are depicted in Fig 6. Walking indicates high relations in all the countries, as shown in the actual physical activities. Basic sports, such as fitness and aerobics, are shown equally in Korea, the USA, and Japan. Also, sports such as yoga appear to be practiced universally and have been adopted as actual physical activities. In Korea, the correlation between golf and yoga is high. That is to say, people enjoying yoga enjoy golf. Notably, the sports facilities, including 467 outdoor and 10,335 indoor golf courses, are equipped to allow people to enjoy golf easily [45]. Interestingly, yoga is enjoyed by many women in Korea [46]. Since yoga can be easily accessed therefore it is highly favored by Korean women, and many people seem to be enjoying golf and yoga simultaneously.

Thirdly, the correlations shown in SNS activities are exhibited in Fig 7. The highest correlations are shown for martial arts and jump rope in Korea, ping-pong and tennis in the USA, and fencing and martial arts in Japan as per the SNS activities of each country. Regarding the connections between sports in each country, the highly connected sports are martial arts and jump rope, with fitness and gymnastics ranking second, and mountain climbing and walking placing third. This finding suggests that a high connection can be seen between highly related sports. However, in the case of the USA, while the highest connection is between ping-pong and tennis, the correlation between billiards and swimming ranking second could not be found. Furthermore, despite the ranking that gives third place to cycling and fitness, fishing and mountain climbing, two seemingly unrelated sports, were given fourth place. The

**Table 9. Ranking of pairs (Top 30—Sport classification).**

| Sport Type | Sport Classification | | Textbook | | | Survey | | | SNS | | |
|---|---|---|---|---|---|---|---|---|---|---|---|
| | | | KO | US | JP | KO | US | JP | KO | US | JP |
| Badminton | Tennis | 1 | 204 | 20 | 39 | 123 | 141 | 139 | 27 | 55 | 113 |
| Ping-pong | Tennis | 2 | 204 | 108 | 17 | 58 | 104 | 66 | 13 | **1** | 95 |
| Badminton | Ping-pong | 3 | 204 | 108 | 39 | **6** | 14 | 15 | 19 | 49 | 89 |
| Aerobic | Dancing | 4 | 204 | 15 | 64 | 123 | **9** | **9** | 15 | **5** | 136 |
| Mountain Climbing | Walking | 5 | 204 | 30 | 64 | 44 | 101 | 94 | **3** | **9** | 12 |
| Aerobics | Gymnastics | 6 | 203 | 108 | 64 | 97 | 154 | 156 | 142 | 51 | 150 |
| Dancing | Yoga | 7 | 170 | 108 | 64 | 123 | 12 | 11 | **8** | 21 | 47 |
| Dancing | Gymnastics | 8 | 151 | 108 | **2** | 123 | 96 | 112 | 50 | 101 | 27 |
| Swimming | Track and Field | 9 | 38 | 108 | 19 | 123 | 248 | 232 | 60 | 30 | 87 |
| Billiards | Bowling | 10 | 204 | 108 | 64 | 85 | 30 | 34 | 17 | 52 | 83 |
| Fitness | Gymnastics | 11 | 93 | 11 | **1** | 42 | 90 | 76 | **2** | 14 | 51 |
| Gymnastics | Martial Arts | 12 | **5** | 108 | **6** | 93 | 209 | 214 | 91 | 81 | 86 |
| Jumprope | Yoga | 13 | 36 | 108 | 64 | 43 | 85 | 80 | 222 | 89 | 211 |
| Dancing | Fitness | 14 | 158 | 106 | **7** | 123 | 32 | 44 | 30 | 40 | 56 |
| Swimming | Yoga | 15 | 41 | 24 | 64 | 123 | 27 | 30 | 89 | 119 | 164 |
| Aerobic | Fitness | 16 | 142 | 49 | 64 | 37 | **3** | **3** | 53 | 28 | 179 |
| Dancing | Martial Arts | 17 | 121 | 108 | **8** | 123 | 178 | 180 | 159 | 103 | 157 |
| Basketball | Football | 18 | **3** | **8** | 26 | 36 | 25 | 21 | 31 | **6** | 97 |
| Fitness | Yoga | 19 | 91 | 28 | 64 | 71 | 11 | 13 | **5** | 18 | 14 |
| Cycling | Mountain Climbing | 20 | 204 | 13 | 64 | 123 | 140 | 143 | 75 | 26 | 118 |
| Mountain Climbing | Yoga | 21 | 204 | 24 | 64 | 91 | 157 | 118 | 194 | 112 | 198 |
| Walking | Yoga | 22 | 41 | 14 | 64 | 20 | **4** | **4** | 158 | 111 | **7** |
| Dancing | Jumprope | 23 | 182 | 108 | 64 | 123 | 107 | 115 | 166 | 199 | 211 |
| Gymnastics | Yoga | 24 | 40 | 35 | 64 | 23 | 62 | 63 | 95 | 36 | 99 |
| Aerobic | Yoga | 25 | 155 | 69 | 64 | 81 | **6** | **6** | 11 | 11 | 178 |
| Baseball | Tennis | 26 | **7** | 36 | 43 | 123 | 227 | 178 | 57 | 46 | **8** |
| Mountain Climbing | Jumprope | 27 | 204 | 75 | 64 | 30 | 98 | 81 | 254 | 144 | 264 |
| Mountain Climbing | Track and Field | 28 | 155 | 108 | 64 | 107 | 212 | 168 | 111 | 41 | 140 |
| Fitness | Jumprope | 29 | 102 | 50 | 64 | 104 | 124 | 127 | 149 | 31 | 203 |
| Gymnastics | Track and field | 30 | **4** | 108 | 15 | 30 | 44 | 35 | 139 | 83 | 107 |

connections between two otherwise unconnected sports are also ranked in the networks of SNS activities in the USA. The result for hiking and fishing in the United States may reflect accessibility issues because these are activities that sometimes require approval for access to participate. In contrast, in Korea and Japan, these sports do not require a permit. Lastly, fencing and martial arts are ranked in its highest position, cycling and walking in the second, and fencing and walking as third highest in Japan. For Japan, it seems that the sports that appear to be highly correlated are ranked high, and the third rank is given to football and tennis.

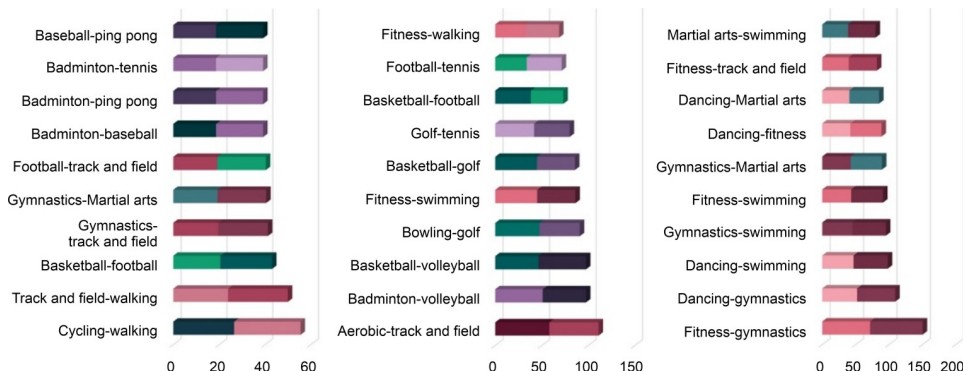

**Fig 5. The K-12 Physical Education System Survey, SNS activity—Top 10 (Korea, USA, Japan).**

Consequently, the connections between different kinds of sports do not have consistent influences on SNS activities. Popular sports, such as basketball, baseball, and football also appeared evenly in the SNS of each country. The result seems to be attributed to the fact that the popularity of sports is related to the SNS activities of the fans [47]. Ultimately, it can be explained that SNS activities are related to both the connections between sports and their popularity.

Finally, the graphs display the top linked sports in the four fields. As shown in Fig 8, the popular sports (i.e., football, basketball, and baseball) are highly connected to various athletics. This result was produced only by survey questionnaires concerning the characteristics of sports types and proved that the existing popular sports have substantial interconnections. This outcome can explain why popular sports today have gained mass popularity. That is to say that popular sports (i.e., football, basketball, and baseball) are considered to have been at the center of the 24 sports not because of external factors like marketing, but because of their unique characteristics. The 41 items selected by the expert group represent the characteristics of each sport. We investigated various characteristics, such as using a large stadium, the participation of several players, and using a goal post. These characteristics appear to differing degrees in popular and unpopular events. A detailed comparison of the characteristics of popular and unpopular sports can elucidate the reasons behind popularity and lack of popularity. Therefore, the comparison can help determine the factors that lead unpopular sports to fail in gaining popularity among the public. Thorough study of the relationship between sports characteristics can allow inference of the reason unpopular sports do not attract public attention. Through this, we can determine approaches to increase public interest in unpopular sports.

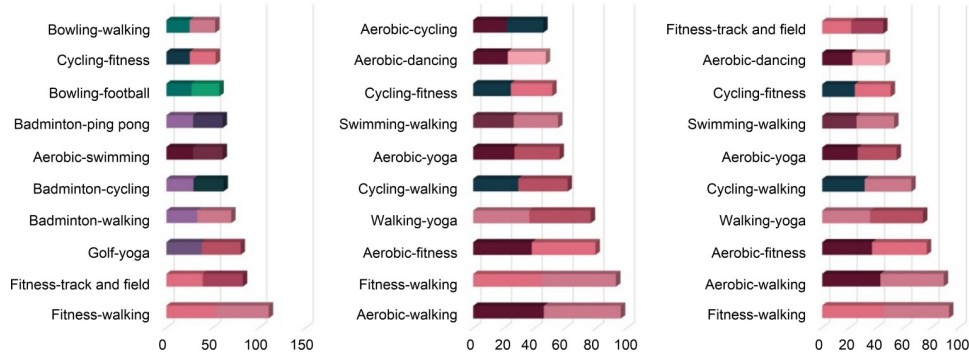

**Fig 6. Survey (actual physical activity)—Top 10 (Korea, USA, Japan).**

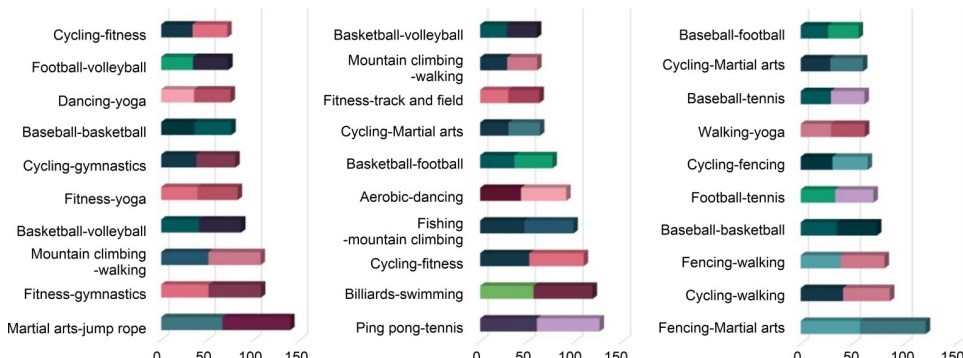

**Fig 7. SNS activities—Top 10 (Korea, USA, Japan).**

As shown in Fig 9, gymnastics (Korea), fitness (USA), and gymnastics (Japan) take the highest place in terms of their connections with other sports regarding educational activities. That is, the principle that emphasizes basic sports seems to be reflected in the curriculum [48]. Looking further into the differences between the countries, martial arts and golf are highly connected in Korea. On the contrary, martial arts and dancing display a high connection in Japan, while swimming and cycling are highly connected in the USA. Korea and Japan have common features in track and field, martial arts, and swimming, while Korea includes golf and jump rope, and Japan includes dancing and fitness. Although they share common features that place great significance on basic sports, there are differences in the sports connected to nationally. It seems that cultural variations have been reflected in the process of organizing the curricula.

Through the survey (actual physical activities), walking displays the highest connection with other sports (Fig 10). Based on the result that fitness, gymnastics, aerobic, and cycling are highly connected, the basic sports are seen to be mostly played as actual physical activities. There is a noticeable result that swimming is preferred in Japan and the USA, but badminton and bowling replace them in Korea. The result appears to be attributed to the fact that Korea has only 1% of swimming pools on a national average, which is very scarce compared to Japan, which reports a penetration rate of swimming pools ranging between 90%–98% for primary schools in each region [49]. It reminds us of the necessity to expand sports facilities to popularize sports.

As seen in Fig 11, cycling in Korea and the USA and walking in Japan are highly connected to SNS activities. SNS activities are different from actual physical activities in that the users mainly post pictures of sports with spectacular subjects such as cycling, martial arts, and fitness. This phenomenon is because SNS activity can reveal ostentatiousness [50]. To upload posts solely to show off makes it equally necessary to take pictures that attract the attention of

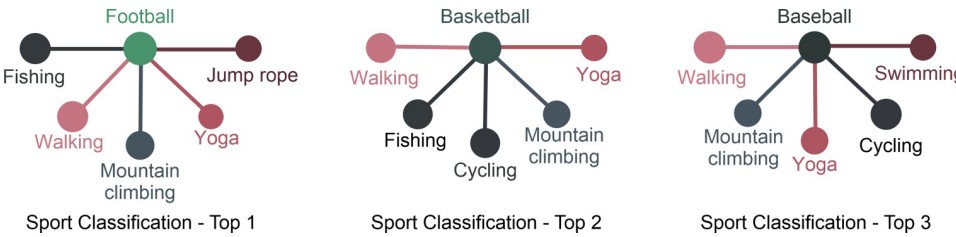

**Fig 8. Highly connected sports (Sport classification).**

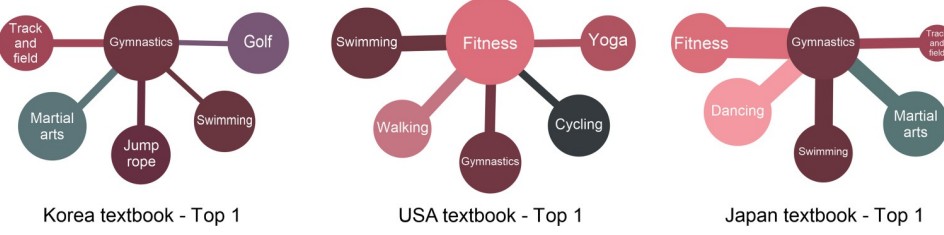

Korea textbook - Top 1          USA textbook - Top 1          Japan textbook - Top 1

**Fig 9. Highly connected sports (Textbook).**

the masses. Since cycling is an outdoor sport, it seems to have high connections. What is special here is that walking has the most connections on SNS in Japan. It seems to reflect the cultural characteristics of Japan, which values everyday triviality.

## Comparison with sport classification

The degree of connection between sports types was compared among the four fields. Textbooks, surveys, and SNS were compared based on the sport classification. Table 10 shows the ranking of sports types in the four fields.

According to the results, football, basketball, and baseball are located in a high degree of connectivity, while fitness, track and field, and gymnastics are located in a low degree of connectivity. Football, basketball, and baseball are the most popular sports. Fitness, track and field, and fitness are basic sporting events. There are many different sports; some are popular and the others are less popular. This evidence can be attributed to characteristics, such as strength, stamina, speed, density, aggression, and team spirit [51]. The ranking of the connection diagrams presented here denotes where each sport is located. Therefore, we compared the links shown in sports classification and the line in textbooks, surveys, and SNS as follows. The number of sports connections was measured in the order of sports types. The graph of the sport classification is prepared on the y-axis and textbooks, surveys, and SNS on the x-axis.

The results show that the rankings of sports covered in textbooks in Korea, the United States, and Japan makes it difficult to find common features (Fig 12). The system that contains the contents of the physical education curriculum is similar in general, but it has national characteristics. Korea has a national-level curriculum, textbooks, and teacher guidance, while the United States presents instruction manuals for teachers in core regional-level curriculum (Lee, 2017 [35]). Japan offers specific details of physical education classes for students by grade (Lee and Koo, 2011 [36]). The results are believed to have been led by the fact that each country has different physical education policies.

A comparison of the survey and sports classification is shown in Fig 13. The results of this survey verify the state of sports that people actually play. The findings highlight that football, walking, cycling, and fitness are found in high places. Volleyball, billiards, bowling, and tennis

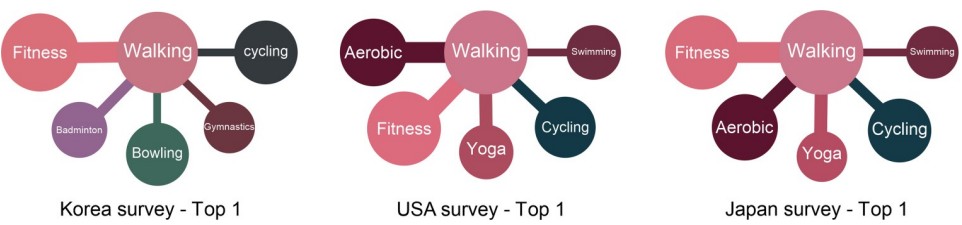

Korea survey - Top 1          USA survey - Top 1          Japan survey - Top 1

**Fig 10. Highly connected sports (Survey).**

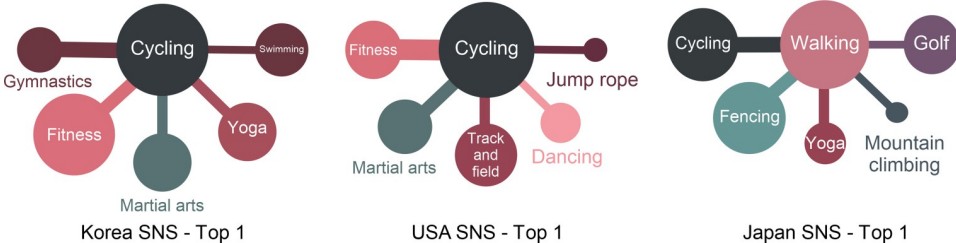

**Fig 11. Highly connected sports (SNS).**

were commonly in low positions. According to a study that analyzed the correlation between demands for sports and national income, basketball, soccer, volleyball, track and field, and martial arts have little correlation with national income. On the other hand, there are also reports that team sports, walking, and fishing are highly correlated with income [52]. As of 2018, the GNI of Korea, the U.S., and Japan have a difference of 30,600 in Korea, 62,850 in the U.S., and 41,340 in Japan. Nevertheless, it appears that no particular difference was detected in physical activity.

A comparison of SNS and sport classifications is shown in Fig 14. SNS results show the status of sports types that people want to show off. According to the results, all three countries show the same position for football, baseball, cycling, fishing, and martial arts.

**Table 10. Ranking of four fields (Sport types linked).**

| Sport Type | Sport Classification | Textbook | | | Survey | | | SNS | | |
|---|---|---|---|---|---|---|---|---|---|---|
| | | KO | US | JP | KO | US | JP | KO | US | JP |
| Football | 1 | 3 | 10 | 6 | 10 | 12 | 11 | 3 | 4 | 5 |
| Basketball | 2 | 5 | 11 | 10 | 16 | 8 | 8 | 6 | 6 | 6 |
| Baseball | 3 | 9 | 18 | 7 | 21 | 21 | 21 | 8 | 11 | 4 |
| Walking | 4 | 19 | 7 | 14 | 1 | 1 | 1 | 10 | 18 | 1 |
| Cycling | 5 | 20 | 5 | 14 | 3 | 4 | 4 | 1 | 1 | 2 |
| Volleyball | 6 | 4 | 4 | 9 | 23 | 23 | 23 | 15 | 15 | 15 |
| Fishing | 7 | 23 | 14 | 14 | 17 | 11 | 12 | 24 | 22 | 21 |
| Billiards | 8 | 23 | 22 | 14 | 12 | 14 | 14 | 21 | 21 | 19 |
| Fencing | 9 | 16 | 22 | 14 | 24 | 24 | 24 | 23 | 24 | 3 |
| Golf | 10 | 6 | 12 | 14 | 9 | 13 | 13 | 4 | 14 | 7 |
| Badminton | 11 | 10 | 13 | 8 | 5 | 10 | 10 | 16 | 20 | 18 |
| Mountain Climbing | 12 | 22 | 9 | 14 | 14 | 18 | 17 | 19 | 9 | 20 |
| Bowling | 13 | 13 | 19 | 14 | 4 | 5 | 5 | 18 | 23 | 22 |
| Ping Pong | 14 | 11 | 22 | 13 | 8 | 16 | 18 | 13 | 17 | 14 |
| Tennis | 15 | 12 | 3 | 12 | 20 | 20 | 20 | 12 | 7 | 8 |
| Swimming | 16 | 7 | 2 | 5 | 15 | 6 | 6 | 11 | 3 | 12 |
| Martial arts | 17 | 2 | 21 | 3 | 19 | 22 | 22 | 9 | 8 | 10 |
| Yoga | 18 | 14 | 8 | 14 | 11 | 7 | 7 | 7 | 10 | 16 |
| Aerobic | 19 | 21 | 15 | 14 | 18 | 3 | 3 | 22 | 16 | 24 |
| Jump Rope | 20 | 8 | 16 | 14 | 13 | 17 | 19 | 20 | 19 | 23 |
| Dancing | 21 | 18 | 20 | 2 | 22 | 9 | 9 | 14 | 13 | 17 |
| Fitness | 22 | 17 | 1 | 4 | 2 | 2 | 2 | 2 | 2 | 11 |
| Track and Field | 23 | 15 | 17 | 11 | 7 | 19 | 16 | 5 | 5 | 9 |
| Gymnastics | 24 | 1 | 6 | 1 | 6 | 15 | 15 | 17 | 12 | 13 |

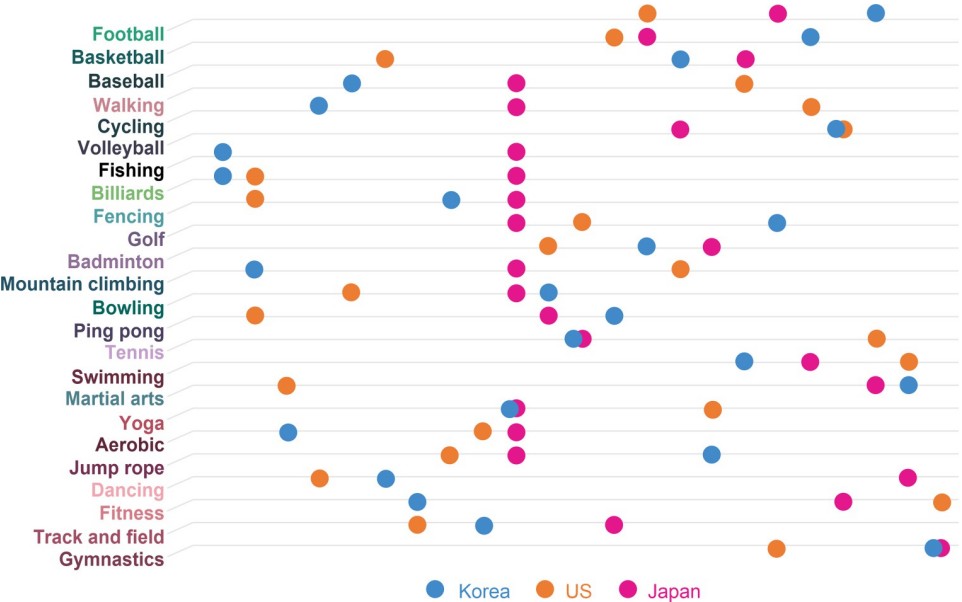

**Fig 12. Comparison with textbooks.**

Instagram users collected in this study's post-SNS phase reported various motivations, such as social interaction, archiving, self-satisfaction, escape, and peeping income [53]. The study found that events, such as football, basketball, cycling, and martial arts were highly displayed in SNS activities among the three countries. The frequent display of football and basketball on SNS seems to be caused by the fact that they have strong fan bases. Cycling and martial arts are shown to have a high degree of connection due to self-examination and self-satisfaction income [54].

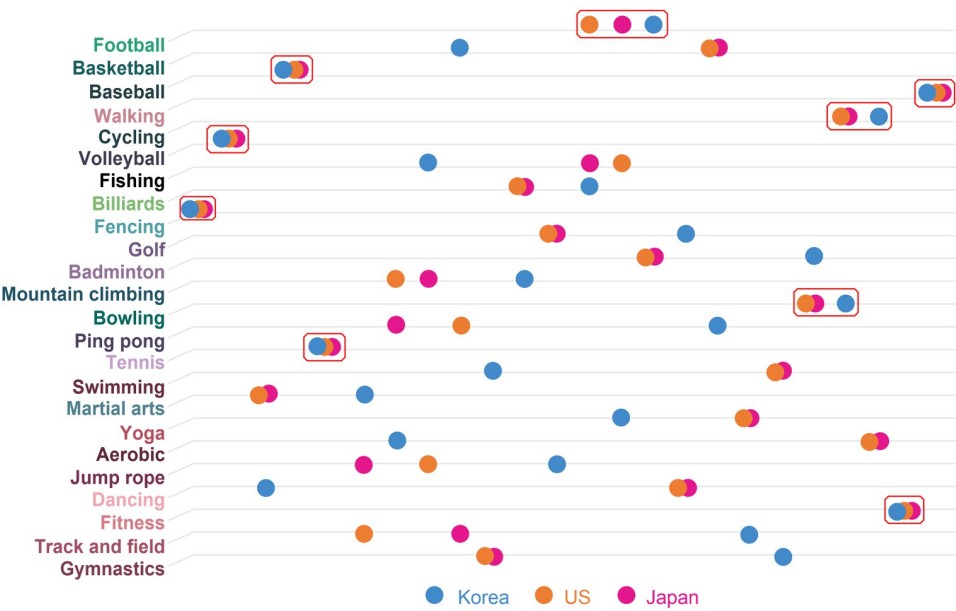

**Fig 13. Comparison with survey.**

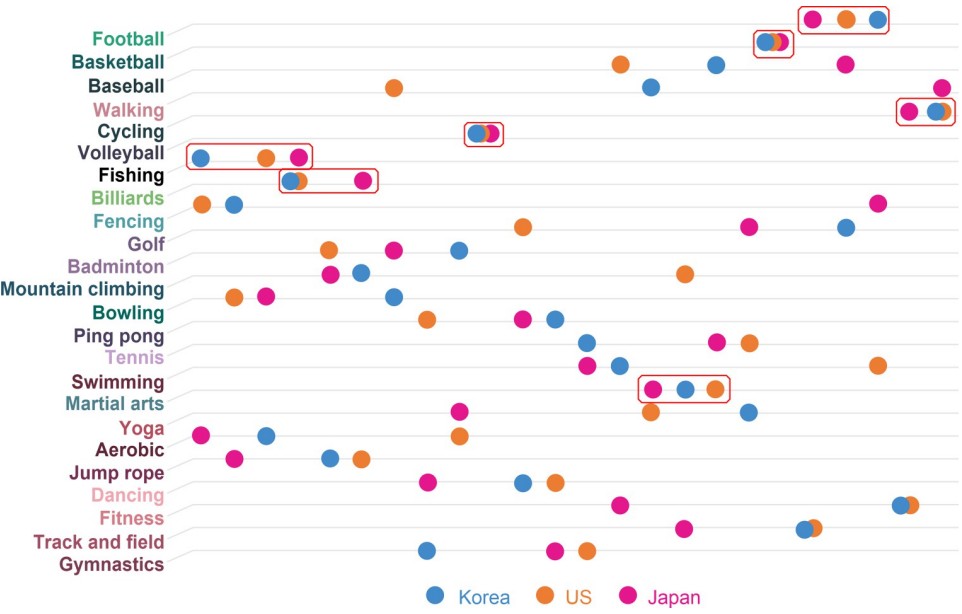

**Fig 14. Comparison with SNS.**

The ranking of the degree of connection according to the characteristics of each sport type has the following meaning. The top sports types of connection are the most popular sports, and the bottom sports types are unpopular sports. Comparing rankings according to the classification of sport types with textbooks, surveys, and social networking sites can be used to identify the location of sport types.

The academic distinction between popular and unpopular sports is not clear, but popular sports can be considered as sports that have been established as professional, activated by their sports club members, or have developed sports infrastructure as per the national base expansion. Moreover, winning medals at various Olympic Games, World Cup tournaments, and Asian Games is meaningful, but the public's perception of unpopular sports is considerably low. This belief is a hindrance to the balanced development of sports, which requires a balanced selection of sports.

Textbooks, surveys, SNS, and comparison of Sport Classification have the following meanings. The diversity among textbooks covers is clearly evident. The results of the survey confirmed the actual activities of sports. These results can be used as basic data to infer the position of a sport type by identifying sport classifications. It can also be used as a reference to select sports to be covered in textbooks.

## Conclusion

This study models the sports types as well as "The K-12 Physical Education System," "Survey (actual physical activities)," and "SNS activities" in Korea, USA, and Japan through the sports networks and analyses them. As a result, four implications are summed up as follows.

First, what was learned from the curriculum and actual physical activities examined through the surveys varies. Second, there are distinct differences between the three countries based on the K-12 Physical Education System, Survey (actual physical activities), and SNS posting activities. Third, SNS posting activities and actual physical activities vary. Fourth, the sports networks on Sport Classification, the K-12 Physical Education System, Survey (actual physical activity), and SNS activities differ significantly.

As identified in the connections between different types of sports, the K-12 physical education system does not influence the actual physical activities. If schools are expected to provide various experiences from different sports, the current curriculum can be appropriate since it already handles many types of sports. However, there can be an opposite opinion suggesting that the curriculum focusing on basic sports is more appropriate in a limited educational environment than in sports that are never played in adulthood. The notable point here is that the students should learn sports to maintain their physical and mental health. To achieve this goal, it is necessary to think more carefully about the structure of the curriculum on the K-12 physical education system, which balances between giving experiences of various sports and teaching the basic sports.

## Author Contributions

**Conceptualization:** Yong-Wook Kim, Jinyoung Han, Minsam Ko.

**Data curation:** Yong-Wook Kim, Kyungtae Jang, Jaewoo Park, Seungyup Lim, Jin-Young Lee.

**Formal analysis:** Yong-Wook Kim, Jinyoung Han, Minsam Ko.

**Funding acquisition:** Yong-Wook Kim, Jinyoung Han, Minsam Ko.

**Investigation:** Yong-Wook Kim, Kyungtae Jang, Jaewoo Park, Seungyup Lim, Jin-Young Lee.

**Methodology:** Yong-Wook Kim, Jinyoung Han, Minsam Ko.

**Project administration:** Yong-Wook Kim, Jinyoung Han.

**Resources:** Yong-Wook Kim, Kyungtae Jang, Jaewoo Park, Seungyup Lim, Jin-Young Lee.

**Software:** Yong-Wook Kim.

**Supervision:** Jinyoung Han, Minsam Ko.

**Validation:** Jinyoung Han, Minsam Ko.

**Visualization:** Yong-Wook Kim, Jinyoung Han.

**Writing – original draft:** Yong-Wook Kim, Jinyoung Han.

**Writing – review & editing:** Yong-Wook Kim, Jinyoung Han, Minsam Ko.

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
