## [Decision Letter · Decision Letter 0]

15 Dec 2021

PONE-D-21-31493The connection to the public's preferred sports analysis and physical education curriculumPLOS ONE

Dear Dr. Ko,

Thank you for submitting your manuscript to PLOS ONE. After careful consideration, we feel that it has merit but does not fully meet PLOS ONE’s publication criteria as it currently stands. Therefore, we invite you to submit a revised version of the manuscript that addresses the points raised during the review process.

We look forward to receiving your revised manuscript.

Kind regards,

Rabiu Muazu Musa, PhD

Academic Editor

PLOS ONE

Journal Requirements:

"This work was supported in part by the Republic of Korea’s Ministry of Education and the National Research Foundation (NRF-2020S1A5B8104091) and by a grant from the Technology Advancement Research Program funded by the Korean Ministry of Land, Infrastructure and Transport (21CTAP- C152247-03)"

"Ministry of Education of the Republic of Korea and the National Research Foundation of Korea (NRF-2020S1A5B8104091; https://www.nrf.re.kr; provided to M.K.);

Technology Advancement Research Program funded by Ministry of Land, Infrastructure and Transport of Korean government.(21CTAP- C152247-03; https://www.kaia.re.kr/; provided to M.K.)

The funders had no role in study design, data collection and analysis, decision to publish, or preparation of the manuscript"

Reviewers' comments:

Reviewer's Responses to Questions

**Comments to the Author**

1. Is the manuscript technically sound, and do the data support the conclusions?

Reviewer #1: Yes

2. Has the statistical analysis been performed appropriately and rigorously? 

Reviewer #1: Yes

3. Have the authors made all data underlying the findings in their manuscript fully available?

Reviewer #1: Yes

4. Is the manuscript presented in an intelligible fashion and written in standard English?

Reviewer #1: Yes

5. Review Comments to the Author

Reviewer #1: Overall comments

This is a unique view of connections between sports that are taught, engaged in, and discussed online. The choice of analysis and the visual components of the data are a wonderful way to demonstrate the multi-dimensional connections and dynamic interactions between components. Very well organized and easy to read. I enjoyed reading about your study and findings.

Comments

Line Number Issue/Suggestion

Abstract

21-22 Purpose stated does not match the data collected – WHY was not investigated, instead, on line 133, the object was to explain the sport network.

Introduction I really enjoyed the introduction. You really catch the interest of the reader and paint a great picture of the purpose of the paper.

109 “also” what do you mean this is in addition to?

110 Confusing subject of the sentence; perhaps, “Therefore, to integrate cooperation into physical education, various sports should be included…”?

111 “two conditions” unclear what two conditions…the two ways curriculum is built or student interest vs. group sports?

129 Please clarify what you mean by “products”. Is this related to products marketed to physical education programs or to the public?

133 Not sure if ‘networks’ should be plural or not

134-135 Could use more of a description at this point about what content you are looking at for the correlations. For instance, cross over of players like in the introduction, movement skills, strategies, popularity, etc. This doesn’t become clear until you get into the methods and results so it would be nice to understand it at this point.

145-146 Great description of the idea of visualizing social networks.

Research Method

179-182 Suggest deleting “Finally, 24 sports were selected” and simply moving the list up to line 177 when you first say that 24 sports were included.

185 How many experts were utilized in this stage of the study?

Additionally, did each expert have to answer all 41 questions for each of the 24 sports? Some more details are needed here to understand the process.

Finally, has this system been used and validated previously? Did you run tests for internal consistency reliability in the experts’ responses?

211 Verb tense � are needs to be ‘were conducted’

211-213 It might be nice to have a total number of survey participants listed here.

Table 4 The results and discussion do not include any mention of job classification, age group, or monthly income – including it here seems irrelevant unless you connect it to the networks

227 Please define ‘web crawling’

228 What are ‘the data’ here? It says Instagram on Table 5, but is this the raw number of posts that were found? Did your team go through and verify the hits or did you simply do a key word search and record the number of results that the search engine provided? Please more clearly describe what the data are.

Results and discussion

285 This seems like a very important finding

289 The use of “Intentionally” indicates you have insight…did you ask about that or is it an assumption of the authors?

312 “unenjoyable”…this wasn’t asked anywhere and needs to be more clear that it is a possible reason as opposed to a fact.

326-327 Could it also be due to the fact that the categories may not have been accurate or reliable? Has the categorizing method been validated? I don’t remember this being noted in the methods.

359 Is yoga considered a sport?

378 An issue of ‘access’ may come in to play such as that in the US, permits are usually needed for fishing and for parking in areas to access mountain climbing.

402 Please include an example of how they ‘can be comprehended’

404-405 Can you provide an example of how the information will lead to mechanisms.

526 Suggest changing the word “learning” as you did not study whether anyone learned things from PE, but you did assess the curriculum that was being taught.

6. PLOS authors have the option to publish the peer review history of their article (what does this mean?). If published, this will include your full peer review and any attached files.

Reviewer #1: No

---

## [Author Response · Author response to Decision Letter 0]

18 Jan 2022

Response to reviewers

Reviewer #1: Overall comments

This is a unique view of connections between sports that are taught, engaged in, and discussed online. The choice of analysis and the visual components of the data are a wonderful way to demonstrate the multi-dimensional connections and dynamic interactions between components. Very well organized and easy to read. I enjoyed reading about your study and findings.

Comments

Line Number Issue/Suggestion

Abstract

21-22 

Purpose stated does not match the data collected – WHY was not investigated, instead, on line 133, the object was to explain the sport network.

[Original]

20-23 

People have their favorite sports type but such preferences tend to be shared for almost a lifetime. How this preference persists is still inconclusive; hence, this study tries to determine why people have different viewpoints on sports. The sports types that people play have been collected through surveys and comparisons between sports networks.

[Changed version]

People have their favorite type of sport, but such preferences tend to be shared for nearly a lifetime. 

How this preference persists is remains inconclusive; hence, this study attempts to determine why people have different viewpoints on sports. 

It is reasonable to infer that these differences arise from differences in culture, occupation, and race. 

Therefore, we collected the following data and conducted research in Korea, the United States, and Japan, countries with various differences.

The types of sports that people play were collected through surveys and comparisons among sports networks.

[Answer]

We have provided additional explanation as to why the data were collected.

Thank you.

Introduction I really enjoyed the introduction. You really catch the interest of the reader and paint a great picture of the purpose of the paper.

[Answer]

Thank you so much.

109 “also” what do you mean this is in addition to?

[Original]

109-110 

It is also difficult to create a sense of cooperation that is the basis of social life.

[Changed version]

It is difficult to create a sense of cooperation that is the basis of social life.

[Answer]

We have removed “also.” 

Thank you.

110 Confusing subject of the sentence; perhaps, 

“Therefore, to integrate cooperation into physical education, various sports should be included…”?

111 “two conditions” unclear what two conditions…the two ways curriculum is built or student interest vs. group sports?

[Original]

109-113 

It is also difficult to create a sense of cooperation that is the basis of social life. 

Therefore, various sports, such as group sports, should be carried out to attract students' interest. 

These two conditions are contrary to each other. 

How these two things are harmoniously distributed leads to either the success or failure of physical education.

[Changed version]

It is difficult to create a sense of cooperation that is the basis of social life. 

Therefore, to integrate cooperation into physical education, various sports, such as group sports, should be utilized to attract students' interest. 

The physical education curriculum should consist of 'Basic exercises' and 'Fun sports' in harmony.

These two conditions are contrary to each other. 

How the two are harmoniously distributed leads to either the success the or failure of physical education.

[Answer]

We have revised the sentence so that readers can easily understand it.

Thank you.

129 Please clarify what you mean by “products”. Is this related to products marketed to physical education programs or to the public?

[Original]

128-131 

In the spirit of economics, many studies examining the marketing effectiveness of product sales have been conducted. 

In deciding a location for a store and displaying products, the relationship between selling products and customers’ preferences has been investigated in some places such as department stores. 

[Changed version]

Various studies have been conducted on the recognition of products and consumers in the marketing research field of public goods.

In determining a location for a store and displaying products, the relationship between selling products and customers’ preferences has been investigated in locations such as department stores. 

[Answer]

We have revised the sentence so that readers can easily understand it.

Thank you.

133 Not sure if ‘networks’ should be plural or not

[Original]

133 

Thus, to explain the sports networks, which is the objective of this study, 

[Changed version]

Thus, to explain the sports network, which is the objective of this study, 

[Answer]

We agree with the reviewer's opinion and have revised this text.

Thank you.

134-135 

Could use more of a description at this point about what content you are looking at for the correlations. 

For instance, cross over of players like in the introduction, 

movement skills, strategies, popularity, etc. 

This doesn’t become clear until you get into the methods 

and results so it would be nice to understand it at this point.

[Original]

134-141

Studies on the analysis of correlations between sports are hardly found except in research that classifies and categorizes the 

characteristics of sports and then determines their positions [26,27]. 

For an accurate analysis of correlations among sports, it is necessary to draw various sports networks, decipher their meanings, and explain them with great insight. It is also imperative to 

adapt to the means of network analysis that thoroughly examines the K-12 Physical 

Education System, as well as the actual physical activity and Simple Notification 

Service (SNS) activities required to conduct such a study.

[Changed version]

Studies on the analysis of correlations among sports are rare except in research that classifies and categorizes the 

characteristics of sports and then determines their positions [26,27]. 

For example, a variety of factors can be correlated, such as popularity, stadium, strategies, and players' movement skills.

For an accurate analysis of correlations among sports, it is necessary to draw various sports networks, decipher their meanings, and explain them with great insight. It is also imperative to 

adapt to a means of network analysis that thoroughly examines the K-12 Physical 

Education System as well as the actual physical activity and Simple Notification 

Service (SNS) activities required to conduct such a study.

[Answer]

We have revised the sentence so that readers can easily understand it.

Thank you.

145-146 Great description of the idea of visualizing social networks.

[Answer]

Thank you so much.

Research Method

179-182 Suggest deleting “Finally, 24 sports were selected” and simply moving the list up to line 177 when you first say that 24 sports were included.

[Original]

175-183

To achieve the goal of this study, data were collected through the process 

described below. A total of 24 sports types were selected as subjects for the analysis 

before data collection. The selection of the sports types is based on “A Survey on the 

Participation in Sports Activities in Korea” conducted through an expert meeting (i.e., 

five scholars from the discipline of sport studies). Finally, 24 sports were selected: 

aerobic, badminton, baseball, basketball, billiards, bowling, cycling, dancing, fencing, 

fishing, fitness, football, golf, gymnastics, martial arts, mountain climbing, ping pong, 

rope skipping, swimming, tennis, track and field, volleyball, walking, and yoga.

[Changed version]

To achieve the goal of this study, data were collected through the process 

described below. A total of 24 sports types were selected as subjects for the analysis 

before data collection: aerobics, badminton, baseball, basketball, billiards, bowling, cycling, dancing, fencing, fishing, fitness, football, golf, gymnastics, martial arts, mountain climbing, ping pong, 

skipping rope, swimming, tennis, track and field, volleyball, walking, and yoga. 

The selection of the sports types was based on “A Survey on the 

Participation in Sports Activities in Korea,” which was conducted through an expert meeting (i.e., five scholars from the discipline of sport studies).

[Answer]

We agree with the reviewer's opinion and have revised the text accordingly.

Thank you.

185 How many experts were utilized in this stage of the study?

[Original]

185-188

Firstly, the “Sport Classification” was collected through expert surveys. The 41 

categories (Table 2) classified by sports characteristics were selected during the experts’ 

meeting, and the data were collected after receiving the response from the experts to 

identify the connections between sports types.

[Changed version]

First, the “Sport Classification” was carried out through expert surveys. 

The experts who participated in the survey were a group of six people with Ph.D degrees in sports.

The expert group held three meetings over the course of two months to select survey items.

The 41 categories (Table 2) classified by sports characteristics were selected during the experts’ meeting, and the data were collected after receiving the response from the experts to 

identify the connections among sports types.

[Answer]

We have revised the sentence so that readers can easily understand it.

Thank you.

Additionally, did each expert have to answer all 41 questions for each of the 24 sports? Some more details are needed here to understand the process.

Finally, has this system been used and validated previously? Did you run tests for internal consistency reliability in the experts’ responses?

[Original]

191-192

Responses were received on a 5-point Likert scale. Also, the distance between the types was measured using Jaccard’s coefficient.

[Changed version]

Responses were provided on a 5-point Likert scale. In addition, the distance between the types was measured using Jaccard’s coefficient.

Cronbach's alpha value was checked to verify the reliability of all 41 questions, and values between 0.822 and 0.929 were found for all questions.

[Answer]

The questionnaire items were selected through three expert discussions in consideration of the characteristics of sports events that do not overlap. In addition, the Cronbach's alpha value was checked to verify the reliability of all 41 questions, and values between 0.822 and 0.929 were found for all questions.

We have added some explanation to this end.

Thank you.

211 Verb tense are needs to be ‘were conducted’

[Original]

211-212

Thirdly, surveys (actual physical activity) in Korea, the USA, and Japan are 

conducted to identify sports types that people actually play.

[Changed version]

Third, surveys (actual physical activity) were conducted in Korea, the USA, and Japan to identify the types of sports that people actually play.

[Answer]

We agree with the reviewer's opinion and have revised this text.

Thank you.

211-213 It might be nice to have a total number of survey participants listed here.

[Original]

211-213

Thirdly, surveys (actual physical activity) in Korea, the USA, and Japan are 

conducted to identify sports types that people actually play. The surveys were 

conducted simultaneously in the three countries for one month in May 2019.

[Changed version]

Third, surveys (actual physical activity) were conducted in Korea, the USA, and Japan to identify the types of sports that people actually play. 

A total of 1,662 people participated in the survey, with 547 people from the United States, 753 people from Korea, and 362 people from Japan participating.

The surveys were conducted simultaneously in the three countries over the course of one month in May 2019.

[Answer]

We agree with the reviewer's opinion and have revised the text.

Thank you.

Table 4 The results and discussion do not include any mention of job classification, age group, or monthly income – including it here seems irrelevant unless you connect it to the networks

[Original]

220-222 

The survey results verified that the number of 

people playing more than two physical activities was 264 (60.55%) in Korea, 505 

(78.54%) in the USA, and 172 (55.66%) in Japan.

[Changed version]

The survey results verified that the number of people participating in more than two physical activities was 264 (60.55%) in Korea, 505 (78.54%) in the USA, and 172 (55.66%) in Japan. The relationship between occupation, age, income level, and sports events collected in the survey will be further analyzed in a follow-up study that has been developed.

[Answer]

We have added an explanation of the purpose of the additional items investigated.

Thank you.

227 Please define ‘web crawling’

228 What are ‘the data’ here? It says Instagram on Table 5, but is this the raw number of posts that were found? Did your team go through and verify the hits or did you simply do a key word search and record the number of results that the search engine provided? Please more clearly describe what the data are.

[Original]

225-228

Lastly, the posting activities of the SNS were used for data analysis, wherein the 

collected posts related to physical activities were uploaded on Instagram. The collection 

was facilitated by web crawling using keywords related to 24 sports types on Instagram. 

The collected data are presented in Table 5. 

[Changed version]

Finally, the posting activities of the SNS were used for data analysis, wherein the 

collected posts related to physical activities were uploaded on Instagram. The collection 

was facilitated by web crawling using keywords related to 24 sports types on Instagram. 

Web crawling is a method of automatically collecting information posted online by creating an Internet bot program.

Through this process, sports types were collected as data from posts uploaded by SNS users.

The collected data are presented in Table 5. 

[Answer]

We have revised the sentence so that readers can easily understand it.

Thank you.

Results and discussion

285 This seems like a very important finding

[Answer]

Thank you so much.

289 The use of “Intentionally” indicates you have insight…did you ask about that or is it an assumption of the authors?

[Answer]

We described this with reference to the opinions of the physical education teachers. According to physical education teachers, they intentionally teach sports that they cannot easily participate in according to the purpose of education. There are various educational purposes, such as cooperation, physical development, and new experiences.

Thank you.

312 “unenjoyable”…this wasn’t asked anywhere and needs to be more clear that it is a possible reason as opposed to a fact.

[Original]

310-312

Hence, the differences between the SNS and actual physical activities can be explained by the trend of posts oriented around boasting privileges, even though they are unenjoyable in actual situations.

[Changed version]

Hence, the difference between SNS and actual physical activity cannot easily be enjoyed in real life; however, posts that boast about privileges tend dominate.

[Answer]

We have revised the sentence so that readers can easily understand it.

Thank you.

326-327 Could it also be due to the fact that the categories may not have been accurate or reliable? Has the categorizing method been validated? I don’t remember this being noted in the methods.

[Original]

326-329

Therefore, it can be thought that the similarities between the sports types have no special meaning. However, as the connections quantify the similarities between sports types, it can be meaningful to select sports without similarities for a curriculum that encourages various physical activities. 

[Changed version]

The similarities between sports types classified by the expert group were confirmed by measuring various characteristics of the sports types. Thus, ping pong and tennis have a similar relationship, as do tennis and badminton, but the results from the three fields were different.

Therefore, it may seem that the similarities between the sports types have no special meaning. However, as the connections quantify the similarities between sports types, it is meaningful to select sports with no similarities for a curriculum that encourages various physical activities. 

[Answer]

We have revised the sentence so that readers can easily understand it.

Thank you.

359 Is yoga considered a sport?

[Answer]

Sports such as yoga and pilates are among the most popular sports events in the East. The size of the industry involved is enormous, and there are many academies in operation.

Thank you.

378 An issue of ‘access’ may come in to play such as that in the US, permits are usually needed for fishing and for parking in areas to access mountain climbing.

[Original]

377-381

Furthermore, despite the ranking that gives third place to cycling and fitness, fishing and mountain climbing, two seemingly unrelated sports, were given fourth place. The connections between two irrelevant sports are also ranked in the networks of SNS activities in the USA.

[Changed version]

Furthermore, despite the ranking that gives third place to cycling and fitness, fishing and mountain climbing, two seemingly unrelated sports, were given fourth place. The connections between two otherwise unconnected sports are also ranked in the networks of SNS activities in the USA.

The result for hiking and fishing in the United States may reflect accessibility issues because these are activities that sometimes require approval for access to participate. In contrast, in Korea and Japan, these sports do not require a permit.

[Answer]

This is information that we were unaware of. Thanks to this comment, we have added clearer explanations for the reader.

Thank you.

402 Please include an example of how they ‘can be comprehended’

[Original]

400-405

If the characteristics of popular and unpopular sports are to be compared in detail, the reasons for popularity and unpopularity can be comprehended. Therefore, the comparison can help find the factors that make unpopular sports fail in gaining popularity among the public. 

If studied thoroughly, these mechanisms can also help design the direction for the development of unpopular sports.

[Changed version]

The 41 items selected by the expert group represent the characteristics of each sport. We investigated various characteristics, such as using a large stadium, the participation of several players, and using a goal post. These characteristics appear to differing degrees in popular and unpopular events.

A detailed comparison of the characteristics of popular and unpopular sports can elucidate the reasons behind popularity and lack of popularity. Therefore, the comparison can help determine the factors that lead unpopular sports to fail in gaining popularity among the public. 

If studied thoroughly, these mechanisms can also facilitate the design of the direction for the development of unpopular sports.

[Answer]

We have revised the sentence so that readers can easily understand it.

Thank you.

404-405 Can you provide an example of how the information will lead to mechanisms.

[Original]

404-405 If studied thoroughly, these mechanisms can also help design the direction for the development of unpopular sports.

[Changed version]

Thorough study of the relationship between sports characteristics can allow inference of the reason unpopular sports do not attract public attention. Through this, we can determine approaches to increase public interest in unpopular sports.

[Answer]

We have revised the sentence so that readers can easily understand it.

Thank you.

526 Suggest changing the word “learning” as you did not study whether anyone learned things from PE, but you did assess the curriculum that was being taught.

[Original]

526-527 

First, the learning from the curriculum and actual physical activities questioned through the surveys vary. 

[Changed version]

First, what was learned from the curriculum and actual physical activities examined through the surveys varies. 

[Answer]

We agree with the reviewer's opinion and have revised the text accordingly.

Thank you.

---

## [Editor Report · Decision Letter 1]

2 Feb 2022

The connection to the public's preferred sports analysis and physical education curriculum

PONE-D-21-31493R1

Dear Dr. Ko,

We’re pleased to inform you that your manuscript has been judged scientifically suitable for publication and will be formally accepted for publication once it meets all outstanding technical requirements.

Kind regards,

Rabiu Muazu Musa, PhD

Academic Editor

PLOS ONE
---

## [Editor Report · Acceptance letter]

21 Feb 2022

PONE-D-21-31493R1 

The connection to the public's preferred sports analysis and physical education curriculum 

Dear Dr. Ko:

I'm pleased to inform you that your manuscript has been deemed suitable for publication in PLOS ONE. Congratulations! Your manuscript is now with our production department. 

Kind regards, 

on behalf of

Dr. Rabiu Muazu Musa 

Academic Editor

PLOS ONE